# Photo-Crosslinked Coumarin-Containing Bis-Urea Amphiphile Hydrogels

**DOI:** 10.3390/gels8100615

**Published:** 2022-09-27

**Authors:** Jie Liu, Xianwen Lou, Maaike J. G. Schotman, Patricia P. Marín San Román, Rint P. Sijbesma

**Affiliations:** 1Institute for Complex Molecular Systems, Department of Chemical Engineering and Chemistry, Eindhoven University of Technology, 5600 MB Eindhoven, The Netherlands; 2Institute for Complex Molecular Systems, Department of Biomedical Engineering, Eindhoven University of Technology, 5600 MB Eindhoven, The Netherlands

**Keywords:** photo-crosslinking, coumarin, bis-urea amphiphile, supramolecular hydrogels

## Abstract

The design of photo-responsive supramolecular hydrogels based on coumarin dimerization and de-dimerization is described. The photo-responsive coumarin unit is chemically incorporated into an oligo(ethylene glycol) (OEG) bis-urea amphiphile that is capable of co-assembling with non-functionalized OEG amphiphile, to form supramolecular fibers. UV light with two different wavelengths (365 nm and 254 nm) is employed to induce a photo-reversible dimerization and de-dimerization process of coumarin units, respectively. The co-assembled solutions could be photo-crosslinked to induce a sol-to-gel transition through dimerization of coumarin with 365 nm UV light, and de-dimerization occurs with 254 nm UV light, to provide a weaker gel. In this system, the mechanical strength of supramolecular hydrogels can be tuned using the irradiation time, providing precise control of gelation in a supramolecular hydrogelator.

## 1. Introduction

Compared to their covalent polymer counterparts, supramolecular hydrogels have definite advantages for mimicking the characteristics of extracellular matrices (ECMs), and therefore, considerable efforts have been taken to develop supramolecular hydrogel systems for applications in tissue engineering and regenerative medicine [1]. Usually, supramolecular hydrogels are intrinsically soft and weak, thereby greatly limiting their applications as ECM mimics, especially as structural elements [2]. In natural ECMs, protein filaments, however, are often crosslinked by dynamic chemical bonds, such as disulfide in fibrillin [3], to enhance the mechanical strength and maintain the highly dynamic nature of ECMs. Inspired by these examples from nature, the use of a covalent crosslinking strategy offers an opportunity to develop supramolecular hydrogels with an enhanced and controllable mechanical strength [4]. Our group previously reported that oligo(ethylene glycol) (OEG) bis-urea amphiphiles assemble into semi-flexible fibers in water via strong hydrogen bonding of urea moieties [5,6,7]. Generally, these supramolecular fibers cannot form hydrogels by themselves, and incorporation of inter-fibrous covalent crosslinkers is required for gelation [8,9,10]. Chemically crosslinking of supramolecular fibers of bis-urea amphiphiles through alkyne-azide cycloaddition yielded strain-stiffening hydrogels that closely mimic the nonlinear mechanics of biological networks [10]. However, the irreversible covalent crosslinking interaction sacrifices the dynamics of supramolecular polymers; thus, the use of reversible dynamic crosslinking rather than the fixed covalent bonds in supramolecular systems is of interest, as a way to capture ECM dynamics [11].

Various chemical crosslinking methods, including Diels–Alder [12], [2 + 2] cycloaddition [13], and boronic ester [14] reactions have been widely exploited to create synthetic polymer hydrogels. Among these methods, UV light-mediated crosslinking, as a facile and versatile strategy, offers many interesting capabilities in polymer networks, such as precise spatial addressability, self-healing, and reversibility [15,16,17]. The coumarin unit, as a photo-reversible reactive group, is ideally suited to serve as the crosslinking functionality in polymer networks. Coumarins undergo photo-dimerization via a [2 + 2] cycloaddition upon irradiation with long-wavelength UV light (>300 nm), and the dimer can be cleaved by short-wavelength UV light (<300 nm) [17,18,19]. The photo-induced dimerization and cleavage reaction represent a reversible photo-switch, which has been used to develop photo-reversible materials for application in photovoltaic devices and tissue engineering [20,21,22].

Herein, we report a novel photo-crosslinked supramolecular hydrogel system, in which coumarin units are incorporated into a OEG bis-urea amphiphile. The coumarin functionalized crosslinker (**Cou-crosslinker**) is combined with non-functionalized OEG bis-urea amphiphile monomers (**P8-10 OMe**) by co-assembly via simple mixing, as indicated in Figure 1. Irradiation with long-wave UV light (365 nm) induced a sol-to-gel transition of supramolecular fiber solution, and the mechanical strength of the resulted hydrogel can be easily tuned by irradiation time. Moreover, the photo-crosslinked interaction is partially reversible with short-wave UV light (254 nm). Such a unique responsiveness of the coumarin group offers an opportunity to control the gelation process and supramolecular material properties, while maintaining the dynamic nature of the supramolecular polymers.

## 2. Results and Discussion

### 2.1. Model Reaction

To test the feasibility of reversible dimerization of coumarin units in water, a model reaction was performed using **Cou-OEG300** as a model compound, which was synthesized in multiple steps (Figure 2). The commercial 7-hydroxycoumarin was first modified with methyl bromoacetate and subsequently converted into free carboxylic acid (**Cou-COOH**) by hydrolysis in a strongly basic environment. The model compound was synthesized by esterification of **Cou-COOH** with oligo(ethylene glycol) (OEG 300) in the presence of thionyl chloride.

When this amphiphilic compound (**Cou-OEG300**) was dispersed at 10 mg/mL in water, a turbid solution was obtained. The turbidity was likely due to the formation of large micelles with coumarin units in the hydrophobic core (Figure 1a). Dimerization of coumarin units leads to the formation of saturated cyclobutane rings through a [2 + 2] cycloaddition, and the progression of the coumarin dimerization was monitored by ^1^H NMR. Upon irradiation with 365 nm UV light from 1 h to 2 h, the integrals of distinct peaks in coumarin units, including the protons of the alkene unit (in position 1, 5 and 6) and the aromatic ring (in position 2, 3 and 4) gradually decreased (Figure 1b). Moreover, numerous new peaks appeared over time, due to the formation of different isomers of coumarin dimers [23,24]. The degree of dimerization of coumarin was calculated from the integral ratio of two protons in positions 6 in the **Cou-OEG300** and 6′ in the dimer. Around 87% of coumarin in the model compound had dimerized after irradiation for 2 h. When the dimerized solution was irradiated with 254 nm UV light for 1 h, around 75% of dimerized coumarin groups were cleaved. These results indicate that the dimerization of coumarin groups in assembled micelle systems occurs under 365 nm UV light irradiation, and the de-dimerization process can be induced by 254 nm UV light. Therefore, the coumarin groups show potential as a reversible crosslinker for supramolecular fibers in water.

The coumarin based photo-crosslinker was obtained through the formation of ester bonds between carboxylic groups from **Cou-COOH** and alcohol groups of the OEG bis-urea amphiphile in a good yield (Figure 3). 

In the ^1^H NMR spectra, all distinct chemical shifts in coumarin modified OEG amphiphile and starting compounds were assigned and are shown in Figure 2. The triplet at 4.58 ppm (position g′) corresponding to -CH_2_- protons next to free alcohol groups in the reactant shifted to 4.26 ppm (position g), while a singlet at 4.73 ppm corresponding to -CH_2_- protons next to the coumarin head in **Cou-COOH** slightly shifted to downfield, to 4.96 ppm (position f), upon functionalization. The other distinct NMR peaks in aromatic coumarin rings did not change after the reaction. The integral values of the protons agree with the expected structure, confirming a high purity.

### 2.2. Solution Preparation and UV-Vis Absorption Measurements

As previously reported, the crosslinker derived from OEG bis-urea amphiphile is capable of co-assembling with non-functional OEG bis-urea amphiphile utilizing a simple mixing method if the two amphiphiles share the same spacing between urea groups in their hydrophobic parts [25,26]. The coumarin functionalized OEG amphiphile crosslinker (**Cou-crosslinker**) did not dissolve in water, probably because the presence of two coumarin end groups caused a significant increase of hydrophobicity of the OEG amphiphile. However, when the **Cou-crosslinker** (at 5% and 10%) was mixed with OEG bis-urea amphiphile, clear solutions were obtained upon sonication in an ice bath, indicating the formation of co-assemblies. Above 10%, a white precipitate or sedimentation in a mixed solution was observed (Appendix A). To probe the photo-responsiveness of coumarin crosslinker in the assembled fiber system, an aqueous solution of OEG bis-urea amphiphile (**P8-10 OMe**) mixed with 10% **Cou-crosslinker** was prepared, with a total concentration of 10 mg/mL. The photo-responsiveness was characterized with UV-Vis spectroscopic analysis. In the mixed solution, a distinct adsorption band at 320 nm was observed, which is the characteristic absorbance for the benzopyrone ring of coumarin [27]. Upon irradiation, the band intensity decreased progressively and gradually leveled off after 30 min of irradiation, as shown in Figure 3a. Upon irradiation with UV light of 254 nm, the absorbance at 320 nm significantly increased and reached its maximum within 5 min, although it did not reach the initial value (Figure 3b). These results indicated that photodissociation is much faster than dimerization, but does not reach full reversion, possibly because that dimerization is not fully reversible [28] or because the coumarin groups degrade in response to UV light [29,30].

### 2.3. Mass Spectroscopy 

To obtain direct evidence of dimerization and de-dimerization in the coumarin based crosslinker, MALDI TOF MS was used to analyze the mixed solution containing **Cou-crosslinker**, before and after irradiation (Figure 4 and Appendix A). In the mass spectrum taken of the solution before irradiation, a peak at 1732 Da was observed for **Cou-crosslinker**, together with multiple peaks of poly-disperse OEG bis-urea amphiphile (Figure 4a and Appendix Aa). Upon irradiation for 10 min at 365 nm, the peak intensity of the **Cou-crosslinker** decreased significantly, and a new peak at 3442 Da appeared at the exact mass of the dimerized product (Figure 4b and Appendix Ab). A small mass peak at 5151 Da was also observed, showing the presence of trimers. A further increase in irradiation time to 30 min led to a progressive decrease of the MS peak intensity of the **Cou-crosslinker**. These results indicate fast coumarin dimerization under UV irradiation (Figure 4c and Appendix Ac). Additionally, upon irradiation with UV (254 nm) for 30 min, the intensity of the dimer peak slightly decreased, while the peak intensity of the **Cou-crosslinker** increased significantly. Notably, the dimerized product was still observed after 30 min 254 nm UV irradiation, indicating an incomplete recovery of coumarin units (Figure 4d and Appendix Ad), in line with the observation in the UV-Vis spectra (Figure 3b).

### 2.4. Dynamic Light Scattering

Dynamic light scattering (DLS) was performed to determine the aggregation behavior before and after 365 nm UV light irradiation. Correlation function curves were recorded on a 1.0 mM mixed solution. Mixing **Cou-crosslinker** with OEG amphiphile (**P8-10 OMe**) provided a single exponential correlation function with a decay time of 8.7 × 10^4^ μs (Figure 5a), which was significantly longer than that of the pure OEG amphiphile solution [7]. It is more likely that a more hydrophobic coumarin based crosslinker leads to the formation of larger aggregates in the mixed solution. Upon irradiation for 30 min, the solution decayed more slowly compared to the non-irradiated solution, and a two-step decay was observed in the correlation functions. These results imply that the aggregates in mixed solutions became larger after irradiation, which is most likely related to inter-fiber crosslinking interaction via coumarin dimerization. 

### 2.5. SAXS and Cryo-TEM Determine Size Change upon Irradiation

To obtain more insights into the structure of the mixed OEG amphiphile system, small-angle X-ray scattering (SAXS) experiments were performed on the mixed solution before and after UV irradiation. The scattering profiles of the OEG amphiphile overlap over the whole *q*-range (Figure 5b), suggesting no structural changes, including the length and rigidity of fibers upon photo-crosslinking [9]. Furthermore, the slope of both scattering profiles in the low *q* regime was close to 1.5, typical for elongated rod-like assemblies. Fitting these profiles with a flexible form factor provided a fiber radius of 3.4 nm, in good agreement with pure OEG amphiphile [7,9]. Subsequently, the morphologies of the mixed solution were also investigated by cryogenic electron microscopy (cryo-TEM). Isolated short fibers were observed in cryo-TEM images before irradiation, and there was no significant difference in the fiber morphology compared to the OEG amphiphile solution. Additionally, when the mixed solution was irradiated with UV light for 30 min, the fiber morphology remained unchanged upon crosslinking, and the assembled fibers showed a tendency to aggregate, possibly indicating the success of photo-crosslinking of fibers by coumarin dimerization (Figure 6).

### 2.6. Photo-Induced Gelation and Its Reversibility

It is evident from the above experiments that the **Cou-crosslinker** underwent a reversible dimerization upon irradiation with different wavelengths of UV light, thus in the next step, photo-induced gelation was explored in a co-assembled mixture of OEG amphiphile with 10% of **Cou-crosslinker**. At a total concentration of 30 mg/mL, the mixed solution remained a viscous sol state, as determined by test tube inversion, while a self-supporting hydrogel was formed upon irradiation with UV light of 365 nm for 30 min. The gel state could not be returned to a sol state with a UV light of 254 nm for 30 min, as shown in Figure 7a; likely because the de-dimerization was incomplete. The mechanical strength of solution or gel in three different states was measured in a rheometer under oscillatory shear. As indicated in Figure 7b, the storage modulus (G′) of the mixed solution was about 16 Pa, exceeding the loss modulus (G′). Upon irradiation with 356 nm UV light, the G′ of crosslinked hydrogel increased fourfold to 66 Pa. Irradiation with 254 nm UV light for 30 min only led to a decrease of G′ to 38 Pa, confirming incomplete dissociation, in line with the observations with UV-Vis in Figure 3c.

The mechanical properties of the photo-crosslinked hydrogels were also studied in situ in the rheometer (Anton Paar), with a home-built 365 nm UV light setup [31]. Before 365 nm UV irradiation, the G′ was approximately 12 Pa, higher than the G″. When the UV lamp was turned on, G′ increased with time, and the increase stopped when the UV light was turned off. When the UV light was turned on again, the G′ started to increase further and reached a plateau value of approximately 60 Pa after 45 min (Figure 7c). These results indicate that crosslinking between the supramolecular fibers by dimerization of coumarin occurs upon UV light irradiation and stops in the absence of UV light, offering an easy way to tune the crosslinking density and gel strength. Unfortunately, the reverse reaction of coumarin dimerization could not be studied in situ, because the used 254 nm UV light source did not fit in the Anton Paar rheometer. 

## 3. Conclusions

A coumarin based photo-responsive crosslinker was synthesized and introduced into non-functionalized OEG bis-urea amphiphile fibers through co-assembly. The coumarin group dimerizes upon irradiation with 365 nm UV light, with a significant decrease in UV absorption of the coumarin groups and mass peak intensity of the coumarin-based crosslinker. The dimerization of the coumarin-based crosslinker leads to a sol-to-gel transition by chemically crosslinking OEG bis-urea amphiphile supramolecular fibers, in which the aggregation of supramolecular fibers is observed. However, the photo-crosslinking between fibers does not change the fiber structure, as shown by the overlap of scattering profiles in SAXS before and after irradiation with 365 nm UV light. Importantly, the modulus of crosslinked supramolecular hydrogels is easily tailored by the irradiation time. The dimerized product of coumarin based crosslinker is partially cleaved with 254 nm UV light, resulting in a weaker gel with a lower storage modulus. The photo-crosslinked hydrogels cannot be reverted to the original sol stage, due to the incomplete reversibility of the photodimerization, as was also observed in UV-Vis. Altogether, this photo-responsiveness of crosslinked supramolecular hydrogels offers the opportunity to gain more control over the gelation process and gel strength in supramolecular systems. In the future, the potential applications in the biomedical field of this crosslinked supramolecular hydrogels will be explored.

## 4. Materials and Methods

### 4.1. Materials

All used materials were commercially available and used without further purification, unless noted otherwise. Deuterated solvents were purchased from Cambridge Isotopes Laboratories. The reaction was carried out under an inert argon atmosphere, and all glassware was dried in an oven before the reaction. Oligo(ethylene glycol) (OEG300), thionyl chloride, and pyridinium p-toluenesulfonate (PPTS) were purchased from Sigma-Aldrich, and 7-hydroxycoumarin and ethyl bromoacetate were purchased from Tokyo Chemical Industry (TCI). The OEG bis-urea amphiphiles, including **P8-10 OMe** and **M8-10 OH** were synthesized as previously reported [7].

### 4.2. Experimental Methods

^1^H NMR and ^13^C NMR spectra were recorded on a 400 MHz NMR (Varian Mercury Vx or Varian 400 MR) operating at 400 MHz for ^1^H NMR and 100 MHz for ^13^C NMR. Chemical shift (δ) is reported in parts per million (ppm) from tetramethylsilane (TMS) or using the resonance of the deuterated solvent as an internal standard. Splitting patterns are labeled as singlet (s), doublet (d), double doublet (dd), triplet (t), quartet (q), and multiplet (m). Matrix-assisted laser desorption/ionization time-of-flight mass spectrometry (MALDI-TOF MS) measurements were carried out on a Autoflex Speed instrument (Bruker, Bermen, Germany), using α-Cyano-4-hydroxycinamic acid (CHCA) as the matrix. Dynamic light scattering (DLS) measurement was carried out on a Melvern Zetasizer Nano S equipped with a He-Ne (633 nm, 4 mV) laser and an Avalanche photodiode detector. Zetasizer software was used to process and analyze the data. UV light-induced dimerization and de-dimerization of coumarin groups were measured using a UV-Vis spectrophotometer (Jasco V-750) at a temperature of 25 °C. 

#### 4.2.1. Synthetic Procedures

Synthesis of 7-carboxymethoxycoumarin (**Cou-COOH**).

Cou-COOH was synthesized as previously reported in Priestley and coworkers [18].

^1^H NMR (400 MHz, *d*_6_-DMSO) δ ppm: 7.99 (d, *J* = 9.5 Hz, 1H), 7.64 (d, *J* = 9.2 Hz, 1H), 7.04–6.89 (m, 2H), 6.31 (d, *J* = 9.5 Hz, 1H), 4.83 (s, 2H).

Synthesis of coumarin functionalized oligo(ethylene glycol) 300 (**Cou-OEG300**).

7-carboxymethoxycoumarin (517 mg, 2.35 mmol) was first refluxed for 3 h in 10 mL of thionyl chloride, providing 7-chlorocarbonylmethoxycoumarin with full conversion, and unreacted thionyl chloride was removed under reduced pressure. Oligo(ethylene glycol) 300 (700 mg, 2.35 mmol) was dissolved in dry tetrahydrofuran (THF), then combined with triethylamine (253 mg, 2.5 mmol) and stirred for 1 h. 7-chlorocarbonylmethoxycoumarin was dissolved in 10 mL THF and added dropwise to mixture solution in an ice bath under argon flow overnight. After filtration of the salt, the resulting product was isolated by silica column chromatography with ethyl acetate/pentane (2:1, *v*/^v^), to provide a 660 mg waxy solid (yield: 58%).

^1^H NMR (400 MHz, CDCl_3_) δ ppm: 7.65 (d, *J* = 9.5 Hz, 1H), 7.41 (d, *J* = 8.6 Hz, 1H), 6.90 (d, *J* = 12.9 Hz, 1H), 6.79 (s, 1H), 6.28 (d, *J* = 9.5 Hz, 1H), 4.73 (s, 2H), 4.44–4.33 (m, 2H), 3.92–3.40 (m, 19H).

^13^C NMR (101 MHz, CDCl_3_) δ ppm: 167.93, 160.87, 160.76, 155.63, 143.23, 129.00, 128.98, 113.76, 113.34, 112.81, 101.81, 101.76, 77.39, 77.07, 76.76, 72.50, 72.47, 70.62, 70.59, 70.56, 70.53, 70.49, 70.29, 68.79, 65.26, 64.55, 61.68.

Synthesis of coumarin functionalized OEG amphiphiles (**Cou-crosslinker**).

First, 74 mg of 7-carboxymethoxycoumarin (**Cou-COOH**, 0.337 mmol) and 48 mg of N, N′-Diisopropylcarbodiimide (DIC, 0.383 mmol) were dissolved in dry THF, and the mixture solution was stirred for 2 h under argon. After removal of THF by vacuum evaporator, 200 mg of HO-PEG-10U6U10 (0.153 mmol) and 9.0 mg of 4-(Dimethyl-amino) pyridinium 4-toluene-sulfonate (DPTS, 0.031 mmol) were added with 5 mL of dry chloroform. The reaction solution was stirred for 48 h at room temperature (other 48 mg DIC was added after 24 h), following washed by brine. Around 92 mg of the final product at a yield of 35% was isolated by silica column chromatography, with MeOH/CHCl_3_ as eluent.

^1^H NMR (400 MHz, *d*_6_-DMSO) δ ppm: 8.00 (d, *J* = 9.5 Hz, 2H), 7.64 (d, *J* = 8.5 Hz, 2H), 7.16 (t, *J* = 5.4 Hz, 2H), 7.04–6.93 (m, 4H), 6.31 (d, *J* = 9.5 Hz, 2H), 5.70 (s, 4H), 4.96 (s, 4H), 4.30–4.19 (m, 4H), 4.09–3.98 (m, 4H), 3.70–3.42 (m, 60H), 2.94 (d, *J* = 6.7 Hz, 13H), 1.22 (s, 47H).

^13^C NMR (100 MHz, *d*_6_-DMSO) δ ppm: 168.68, 161.11, 160.67, 158.61, 156.62, 155.60, 144.69, 130.00, 113.37, 113.14, 102.01, 79.59, 70.22, 69.34, 68.58, 65.39, 64.50, 64.05, 63.45, 40.66, 40.52, 40.31, 40.10, 39.89, 39.68, 39.47, 39.26, 30.48, 29.84, 29.46, 29.42, 29.27, 29.19, 26.85, 26.70, 26.56.

M/z (Measured, MALDI TOF MS): 1731.9 [M + Na]^+1^, calcd for C_84_H_136_N_6_O_30_: 1708.93.

#### 4.2.2. Sample Preparation Method

Samples were prepared by weighting OEG bis-urea amphiphile (**P8-10 OMe**) alone with coumarin based crosslinker (5, 10, and 20%, respectively) in the solid state, followed by the addition of ca.300 μL of chloroform. After solvent evaporation at room temperature, the remaining solid was subsequently re-dissolved in Mili-Q water with the assistance of several cycles of sonication (30 min), and vortexed (1 min) until a full dissolution of the solid material. The assembled structures in the mixed solution were allowed to grow for at least an additional 24 h in the dark environment before the following measurement.

For UV-Vis adsorption, MALDI TOF MS, and DLS measurement, the mixed solution (10 mg/mL) was transferred to quartz cuvettes (1 × 1 cm) followed by irradiation with long-wave UV light (365 nm, UVA lamp) or with short-wave UV light (254 nm, UVC lamp) under continuous stirring using a Luzchem photoreactor (Model LCZ 4V) equipped with an output power of 7.2 W. The irradiation time was varied from 5 min to 60 min.

#### 4.2.3. Cryogenic Transmission Electron Microscopy (Cryo-TEM)

Samples for cryogenic transmission electron microscopy (cryo-TEM) were prepared in an automated vitrification robot (Vitrobot^TM^ Mark III, FEI) at room temperature and a relative humidity >95%. Then, 3 µL of samples were applied on a Quantifoil grids (R 2/2, Electron Microscopy Sciences) and Lacey grids (LC200-CU, Electron Microscopy Sciences), which were glow discharged prior to use (Cressington 208 carbon coater operation at 5 mA for 40 s). Subsequently, excess liquid was blotted away using filter paper for 3 s at −3 mm and vitrified in liquid ethane. The samples were examined on a FEI-Titan TEM equipped with a field emission gun operating at 300 kV. A post-GIF (Gatan imaging filter) 2 × 2 Gatan CCD camera was used for the recording of the images. Micrographs were taken at low dose conditions, using a defocus setting of −10 μm at 25 k magnification or defocus setting of −40 μm at 6.5 k magnification. The software ImageJ (ImageJ 1.53j, National Institutes of Heath, Bethesda, MD, USA) was used for image analysis.

#### 4.2.4. Small-Angle X-ray Scattering (SAXS) 

Small-angle X-ray scattering (SAXS) profiles were recorded on a SAXLAB GANESHA 300 XL SAXS equipped with a GeniX 3D Cu Ultra-Low Divergence microfocus sealed tube source, producing X-rays with a wavelength λ = 1.54 Å at a flux of 1 × 10^8^ ph/s and a Pilatus 300 K silicon pixel detector, with 487 × 619 pixels of 172 × 172 μm^2^ in size, placed at three samples-to-detector distances of 113, 713, and 1513 nm, respectively, to cover a *q*-range of 0.1 ≤ *q* ≤ 4.0 nm^−1^ with *q* = 4π/λ (sinθ/2). The two-dimensional images were averaged to obtain intensity I(q) vs. q profiles and calibrated to absolute scale using Mili-Q water as a reference; standard data reduction procedures, i.e., subtraction of the empty capillary and the solvent contribution, were applied. The samples were prepared at a concentration of 10 mg/mL in Mili-Q water and held in 2-mm quartz capillaries. Small-angle X-ray scattering experiments were performed at 20 °C.

#### 4.2.5. Rheology

The mechanical properties of the photo-crosslinked hydrogels were evaluated on an oscillatory rheometer: TA Discovery HR-3 (TA Instruments, New Castle, DE, USA), equipped with a 5 mm stainless steel sand-blasted plate-plate geometry, to prevent sample slippage in a temperature controller. The temperature was fixed to 20 °C, and 60 μL of the mixed solutions was loaded on the rheometer, with three different treatments:without UV irradiation.irradiated by 365 nm UV light for 30 min.initially irradiated by 365 nm UV light for 30 min, followed by irradiation of 254 nm UV light for 30 min.

A fixed plate-to-plate gap of 1000 µm was used, and mineral oil was placed around the sample to minimize evaporation. Time-dependent measurement was first performed under an oscillatory strain of 1.0% and an angular frequency of 1.0 Hz for 3 h. The frequency sweep was conducted under a fixed amplitude of 1.0%, followed by a strain sweep with a fixed angular frequency of 1.0 Hz. 

For time-dependent in situ rheological measurement, 200 μL of the mixed solutions was loaded with a plate-plate rheometer (Anton Paar, MCR 501), equipped with a custom-made UV-curing setup (Hönle Group, Gilching, Germany) [31]. The bottom plate was made of quartz and the top plate was stainless-steel sand-blasted, with a diameter of 25 mm. The measurement gap was 200 mm. The mixed solution was irradiated by UV light (Bluepoint LED eco, wavelength of 365 nm, Hönle Group) with an intensity of 7.2 mW/cm^2^, and the time-dependent measurement was conducted under an oscillatory strain of 1.0% and an angular frequency of 1.0 Hz. The irradiation sequences of 365 nm UV light were performed on the mixed solution by turning on and off the UV light, to investigate the effect of pulsed irradiation.

## Data Availability

The datasets generated for this study are available on request from the corresponding author.

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
