# Peer review of "Photo-Crosslinked Coumarin-Containing Bis-Urea Amphiphile Hydrogels"

_gels, 2022, doi:10.3390/gels8100615_

Round 1

Reviewer 1 Report

In this paper, the authors designed and synthesized photo-responsive supramolecular hydrogels based on coumarin, dimerization and de-dimerization were regulated by UV light with two different wavelengths (365 nm and 254 nm). This research is well established and conducted, so this reviewer recommend it could be published in Gels after minor revision.  

Comments:

1.      The authors should number the compounds and polymers in the figures/schemes and quote them in the body text to improve the readability.

2.     In figure 1b, the authors mentioned around 76% and 92% of coumarin in the model compound had dimerized after 365 nm UV light irradiation for 1 and 2 h, but the integral area looks not that big of a difference between 5 and 5’ in this 1H NMR spectrum.

3.     In figure 2c, one peak around 5.6 ppm is labeled as “H from urea unit”, but the urea unit is not shown in this structure.

4.     Did the author conduct the coumarin-based crosslinker’s reversibility study of UV-induced dimerization monitored by 1H NMR? Like the model reaction in figure 1b.

5.     Page 5, line 155, “indicating an incomplete recovery of coumarin units”, is there a specific recovery ratio?

6.     In Figure 6, it could be better to add the TEM after UV light irradiation (254 nm, 30 min).

Author Response

Comments:

  1. The authors should number the compounds and polymers in the figures/schemes and quote them in the body text to improve the readability.

Thanks for your nice suggestions. We did not number the compounds and polymers because the numbers and letters were used in labelling protons in NMR spectra (Figure 1 and 2). Instead, we highlighted their abbreviated names in the manuscript to improve the readability.

  1. In figure 1b, the authors mentioned around 76% and 92% of coumarin in the model compound had dimerized after 365 nm UV light irradiation for 1 and 2 h, but the integral area looks not that big of a difference between 5 and 5’ in this 1H NMR spectrum.

We double-checked the raw data of all NMR spectra, and indeed, the protons in positions 6 and 6’ should be selected to calculate the conversion.  

  1. In figure 2c, one peak around 5.6 ppm is labeled as “H from urea unit”, but the urea unit is not shown in this structure.

We have now indicated protons from urea units in Figure 2.

  1. Did the author conduct the coumarin-based crosslinker’s reversibility study of UV-induced dimerization monitored by 1H NMR? Like the model reaction in figure 1b.

Indeed, we have tried to monitor dimerization process by 1H NMR spectra, but most of protons in coumarin ring and -CH2- were not observed in D2O, due to association and concomitant low mobility in the supramolecular assemblies.

  1. Page 5, line 155, “indicating an incomplete recovery of coumarin units”, is there a specific recovery ratio?

We cannot provide a specific recovery ratio because MALDI is not a good method for quantitation. In spite of this, we can confidently say that the recovery of coumarin units was incomplete after UV irradiation at 254 nm for 30 min because the dimerized product was still detected.

  1. In Figure 6, it could be better to add the TEM after UV light irradiation (254 nm, 30 min).

Indeed, there was not the TEM image for the sample after UV light irradiation (254 nm). It is because we did not see significantly difference between non-irradiated sample and 365 nm-irradiated sample. The conclusion from the TEM results is that fiber morphology is retained upon irradiation.

  1. As the authors mentioned, "Coumarin functionalized OEG amphiphile crosslinker did not dissolve in water", did the author test the coumarin-based newly synthesized photocrosslinking with any water-soluble molecule as co-assembly? Plus water insoluble cross-linker makes the use less favorable for biomedical applications.

We did not test the coumarin crosslinker with other water-soluble compounds. The co-assembly usually occurs in molecules shared similar molecular structures, and this approach is widely used in supramolecular chemistry. Coumarin functionalized crosslinker has the same hydrophobic core and urea moieties as water-soluble OEG amphiphiles, and mixing them together results in co-assembly and solubilization of the coumarin crosslinker. Therefore, there is no impediment to its use, also in potential biomedical application.

Here are some papers for co-assembly:

(1) Pal, A.; Karthikeyan, S.; Sijbesma, R. P. Coexisting Hydrophobic Compartments through Self-Sorting in Rod-like Micelles of Bisurea Bolaamphiphiles. J. Am. Chem. Soc. 2010, 132 (23), 7842–7843. https://doi.org/10.1021/ja101872x.

(2) Mellot, G., Guigner, J. M., Jestin, J., Bouteiller, L., Stoffelbach, F., & Rieger, J. (2018). Bisurea-functionalized RAFT agent: A straightforward and versatile tool toward the preparation of supramolecular cylindrical nanostructures in water. Macromolecules, 51(24), 10214-10222.

  1. Secondly, "However, when the coumarin-based crosslinker (at 5% and 10 %) was mixed with OEG bis-urea amphiphile, clear solutions were obtained upon sonication in an ice bath, indicating the formation of co-assemblies." Usually, crosslinker concentration below 1% is fine to cause the crosslinking, how can authors justify the very high concentration (5, 10%) of coumarin-based crosslinker used in this paper?

Mixing 5% crosslinker with OEG bis-urea amphiphile cannot lead to gelation after UV irradiation (365 nm), therefore, we had to further increase the coumarin crosslinker concentration to give a fast gelation, which have potential for biomedical application.

  1. Authors' conclusion about the clear solution obtained by using an aqueous solution of OEG bis-urea amphiphile (P8-10 OMe) mixed with a 10% coumarin-based crosslinker makes the use of such a cross-linker specific for OEG bis-urea amphiphile?

The co-assembly is based on molecular recognition, and in our case to make use of matching bisurea spacers, which is described in the paper below:

Pal, A.; Karthikeyan, S.; Sijbesma, R. P. Coexisting Hydrophobic Compartments through Self-Sorting in Rod-like Micelles of Bisurea Bolaamphiphiles. J. Am. Chem. Soc. 2010, 132 (23), 7842–7843. https://doi.org/10.1021/ja101872x.

  1. TEM image (Figure 6) can not clearly demonstrate the fiber network-like morphology.

Indeed, the TEM images is not of a network, but just the fibers in dilute solution. The aim of this experiment is to demonstrate that the fiber morphology does not change significantly upon crosslinking. The aim is not to show a network, which is very difficult with TEM. We have rephased our sentence:

 When the mixed solution was irradiated by UV light for 30 min, the fiber morphology remains unchanged upon crosslinking, and the assembled fibers appeared to aggregate.

  1. Application is missing in this manuscript, It would be better if authors could provide some findings or insights about the application of coumarin-based crosslinker or photo-crosslinked coumarin-containing bis-urea amphiphile hydrogels.

The aim of this paper is to explore the possibility of facile crosslinking in bis-urea amphiphiles, and the applications of these bolaamphiphiles as cell culture substrates is under investigation. Also, we have added some prospective sentences in the end of the conclusion part.

Reviewer 2 Report

Based on coumarin dimerization and de-dimerization events, the authors have created photo-responsive supramolecular hydrogels. This manuscript features deep structural characterization. However, any findings or insights about the applications of these novel hydrogels would be intriguing and appealing.

1. As the authors mentioned, "Coumarin functionalized OEG amphiphile crosslinker did not dissolve in water", did the author test the coumarin-based newly synthesized photocrosslinking with any water-soluble molecule as co-assembly? Plus water insoluble cross-linker makes the use less favorable for biomedical applications.
2. Secondly, "However, when the coumarin-based crosslinker (at 5% and 10 %) was mixed with OEG bis-urea amphiphile, clear solutions were obtained upon sonication in an ice bath, indicating the formation of co-assemblies." Usually, crosslinker concentration below 1% is fine to cause the crosslinking, how can authors justify the very high concentration (5, 10%) of coumarin-based crosslinker used in this paper?
3. Authors' conclusion about the clear solution obtained by using an aqueous solution of OEG bis-urea amphiphile (P8-10 OMe) mixed with a 10% coumarin-based crosslinker makes the use of such a cross-linker specific for OEG bis-urea amphiphile?
4. TEM image (Figure 6) can not clearly demonstrate the fiber network-like morphology.
5. Application is missing in this manuscript, It would be better if authors could provide some findings or insights about the application of coumarin-based crosslinker or photo or photo-crosslinked coumarin-containing bis-urea amphiphile 2 hydrogels.

Author Response

(The authors gave the same response as above.)

Reviewer 3 Report

I thought your study is well described, planned, and written. The scientific methods and arguments are convincing with a high quality of presentation.

With your quality manuscript, I believe scientists will have a brighter future in the application of a simple photo-crosslinked supramolecular hydrogel in tissue engineering or regenerative medicine. Kudos to the group. 

Author Response

(The authors gave the same response as above.)
